# Return on Investment (ROI) and Development of a Workplace Disability Management Program in a Hospital—A Pilot Evaluation Study

**DOI:** 10.3390/ijerph17218084

**Published:** 2020-11-02

**Authors:** Vincenzo Camisa, Francesco Gilardi, Eugenio Di Brino, Annapaola Santoro, Maria Rosaria Vinci, Serena Sannino, Natalia Bianchi, Valentina Mesolella, Nadia Macina, Michela Focarelli, Rita Brugaletta, Massimiliano Raponi, Livia Ferri, Americo Cicchetti, Nicola Magnavita, Salvatore Zaffina

**Affiliations:** 1Health Directorate, Occupational Medicine Service, Bambino Gesù Children’s Hospital IRCCS, 00165 Rome, Italy; vincenzo.camisa@opbg.net (V.C.); francesco.gilardi@opbg.net (F.G.); annapaola.santoro@opbg.net (A.S.); mariarosaria.vinci@opbg.net (M.R.V.); rita.brugaletta@opbg.net (R.B.); 2Post-Graduate School of Occupational Health, Catholic University of Sacred Heart, 00168 Rome, Italy; livia.ferri@hotmail.it (L.F.); nicola.magnavita@unicatt.it (N.M.); 3Graduate School of Health Economics and Management, Catholic University of Sacred Heart (ALTEMS), 00168 Rome, Italy; eugenio.dibrino@unicatt.it (E.D.B.); americo.cicchetti@unicatt.it (A.C.); 4Health Directorate, Bambino Gesù Children’s Hospital IRCCS, 00165 Rome, Italy; serena.sannino@opbg.net (S.S.); massimiliano.raponi@opbg.net (M.R.); 5Nursing and Health Allied Professionals Service, Bambino Gesù Children’s Hospital IRCCS, 00165 Rome, Italy; natalia.bianchi@opbg.net; 6Human Resources Directorate, Bambino Gesù Children’s Hospital IRCCS, 00165 Rome, Italy; valentina.mesolella@opbg.net (V.M.); nadiamaria.macina@opbg.net (N.M.); michela.focarelli@opbg.net (M.F.); 7Department of Woman, Child & Public Health, Gemelli Policlinic Foundation IRCCS, 00168 Rome, Italy

**Keywords:** disability management, workplace health promotion, workplace wellbeing, healthcare workers, absenteeism, return on investment, break-even analysis, return to productivity

## Abstract

The progressive ageing of the working population and the increase in related chronic diseases tend to affect working capacity. The aim of this study was to evaluate a Workplace Disability Management Program (WDMP) within a pediatric hospital. Absenteeism due to healthcare workers’ (HCWs) pre- and post- WDMP and the related costs were used for the program evaluation. The Return on Investment (ROI), the Break-Even Analysis (BEA) and the value of the average annual productivity of HCWs who took advantage of the Disability Management (DM) interventions to assess the economic impact of the program, were also used. The HCWs enrolled in the program were 131 (approximately 4% of hospital staff), of which 89.7% females and with an average age of 50.4 years (SD ± 8.99). Sick leave days of the HCWs involved decreased by 66.6% in the year following the end of WDMP compared to the previous one (*p* < 0.001). The total estimated cost reduction of absenteeism is 427,896€ over a year. ROI was equal to 27.66€. BEA indicated that the break-even point was reached by implementing the program on 3.27 HCWs. The program evaluation demonstrated the particular effectiveness of the implemented WDMP model, acting positively on the variables that affect productivity and the limitation to work.

## 1. Introduction

The progressive ageing of the population and the increase in related chronic diseases tend to affect healthcare workers’ (HCWs) levels of autonomy and working capacity [1,2,3,4,5]. Currently in Europe, there are twice as many HCWs aged over 50 as those under 25. Additionally, 100 million citizens suffer from musculoskeletal disorders and diseases [6], responsible for more than 50% of absences from work, unsuitability, permanent limitations and incapacity for work [7]. The only costs attributable to these diseases among the elderly working population in Europe are estimated to be higher than 2% of GDP [8,9].

In relation to this scenario worldwide, for about 40 years, Workplace Disability Management Program (WDMP) models have been developed with the aim of matching the productive interests of the company with the needs of disabled or sick workers to find a job that satisfies them, not only in economic terms but also on a human and social level [10]. WDMPs intend to offer support to employees suffering from chronic illnesses or disabilities in order to optimize their professional performance, limiting the negative effects of the disease on their activity, through a global, coherent and progressive approach implemented by the company.

WDMP, in fact, has been established for the multidisciplinary management of the needs of HCWs suffering from mental or physical disabilities. Disability Management (DM) intervention was defined as a systematic and constructive method to ensure job-retention and job-reintegration in competitive employment for individuals with (temporary) disabilities [11]. Through a series of diversified and individualized paths, WDMP strategies help workers to: (a) prevent disease and disability (Disability Prevention (DP)), (b) stay at work (Stay at work (SAW)) or (c) return to work after an absence as quickly and safely as possible (Return to work (RTW)) [11,12,13,14,15,16].

The term DM refers to a model theorized in the 1980s, initially developed in North America and Europe [11,12,13,14,15,16], which today is still poorly applied in Italy. On an international level, DM is widely used in the public field, such as, for example, in the Canadian government, where this method is implemented to prevent and manage absences from work due to illness or injury, with tools such as sick leave and planning of benefits and adjustments of professional duties when returning to work [17,18,19].

Currently, several companies all over the world apply WDMP programs with a high variability of methods difficult to reconduct to useful models for drawing-up operational protocols based on efficacy evidence [11,12,13,14,15,16,20,21,22,23,24,25,26,27,28,29,30,31,32,33]. The main DM models also determined by the socio-cultural contexts and the work methodologies in which they are developed, are mainly attributable to SAW, DP and RTW approaches (the most common and based on the contact with HCWs with long periods of illness-related absences from work) in many interconnected and integrated situations [11,12,13,14,15,16]. Most of the interventions are related to RTW programs for employees who have to return to work following a disabling event. This reinforces the concept that DM is still connected more with the response to a problematic situation than to a preventive action [34]. In Europe, Australia and Asia, for example, in addition to studies on the role of RTW [35,36,37], stress [38,39] and vocational rehabilitation [37,40], also budget studies have been carried out [41] on specific management of chronic diseases [42], usefulness of economic incentives for workers [43] and role of communication between the various medical figures involved in patient-worker management [44].

Conversely, in North America, the following is privileged: evaluation of the investors’ role in DM programs [45,46,47,48,49], analyzes on how to reduce the costs related to absenteeism through DM programs [25,50,51], surveys in which workers themselves express their opinion on the DM activities foreseen by the company [45,48,49,52,53,54] and, finally, indications on the best practices to be included in the DM programs [55,56]. In Italy more recent experiences have been gained [14,15] also in relation to a production reality which is 80% made up of small and medium-sized companies with less than 15 employees without the obligation to hire disabled staff, as required by the Law 68/99. A review of these Italian experiences has recently highlighted the common characteristics [14] relating to the Italian scenario by identifying 4 main trends: recruiting and hiring area [57,58,59,60,61,62], RTW and job maintenance field [63,64], organization of smart working and tele-training with the aim of introducing workers with disabilities into the company [65] and study for the introduction of young graduates with disabilities into the working field [66]. At present, the reality of DM in Italy seems to be almost entirely focused only on the pre-hiring phase.

In recent years, several systematic reviews [11,12,13,14,15,16,20,21,22,23,24,25,26,27,28,29,30,31,32,33] tried to classify and evaluate in their effectiveness the various programs adopted. In particular, Gensby et al. [13] have developed a taxonomy of the existing DMP models and the services they provide which divides the interventions present in each WDMP into 3 levels (personal/individual, organizational, systemic), each divided into the two phases prior to and subsequent to RTW. In fact, the most complete WDMP is the one that provides the level of personal intervention (early contact and structuring of work changes in the pre-return phase, follow-up of physical conditions and possibility of reduced or flexible hours in the post-return), organizational (coordinating role of the case manager, identification of activities, tasks and costs related to the job and worker’s replacement during the absence period; subsequently, follow-up of the worker’s reintegration in the activities (paying attention to the working well-being of the whole WG) and systemic (coordination of the insurance parties and the health figures connected to the rehabilitation before the RTW and, subsequently, the follow-up). The analyses reported in recent systematic reviews [11,12,13,14,15,16] regarding studies on the effectiveness of WDMP programs [67,68,69,70,71,72,73,74,75,76,77], present different approaches and experiences that not only present several limitations in performing the surveys but mainly a scarcity of evaluations made on the basis of sets of validated indicators, such as illness-related absences and measurement of job satisfaction, physical and mental health [11,12,13,14,15,16,78,79]. The conclusions of these reviews effectively show the difficulty of identifying evidence of a unique DM concept that is still nebulous and defining clear indications and key elements of the various WDMP assessed. Furthermore, some interesting differences are presented about the various kinds of disability determined by physical (especially connected to musculoskeletal disorders or cancer) or mental health conditions [27,28,29,30,31,32,33]. Additionally, even rarer are the studies that define an economic evaluation of a WDMP impact, especially through the enhancement of savings obtained by a reduction in absenteeism [21,22,80,81].

This pilot study aims at evaluating the effectiveness of a WDMP implemented since 2017 within an Italian pediatric hospital.

## 2. Materials and Methods

### 2.1. Hospital WDMP

In 2017, the hospital management started a WDMP program as part of the three-year strategic plan of the company’s Workplace Health Promotion (WHP). The program aimed at reducing the impact of a possible disability (deriving from disability, illness or accident) on the individual’s working capacity. In this context, the distribution of work restrictions by risk factor highlights how the main causes are the ergonomic factors and the organizational risks, on which the main preventive actions have been focused. In fact, the improvement actions have concentrated on limiting the risk from biomechanical overload/manual patient movement (MPM) and on the activities necessary for the organizational well-being. Furthermore, in 2011 an ad hoc Working Group (WG) was set-up with the aim of managing of the so-called “difficult suitability,” thus replacing the cases with limitation and, at the same time, identifying the “full” suitability for the assigned task. The WG, coordinated by the Head of Occupational Medicine, is constituted by competent physicians, members of Human Resources (HR) Department and Health Directorate; it meets periodically on a monthly basis. There are three main reasons for entering the HCWs in the program: intra or extra work psychological distress (work related stress, family problems); musculoskeletal disorders; other pathologies, especially neoplastic diseases. These are the main reasons that impact on the work capacity of the HCW who very often manifests discomfort, an altered health condition and/or is often absent from work. The paths that are proposed by the WG are personalized and mainly concern four types of intervention:Internal reorganization of work duties, keeping destination and job assignment unchanged.Proposal to the company management for environmental, ergonomic, structural and technological improvement measures.Paths of psychological support and/or health promotion interventions.Change of working destination or, ultimately, change of job.

The assessment of each case is carried out during the WG meetings and the decisions on the adequate intervention to be implemented or changed along the way are discussed and agreed with the HCW and his/her supervisors. The path can last several months, on average. It is closed when the criticalities detected at the entrance have found the appropriate solution. The various steps of the path are ratified within the monthly WG meetings.

The supervisors and staff responsible of the wards and departments were involved in all the phases of the program from the evaluation of the case to the decision regarding the objective of the personal program of the HCWs and, necessarily in the phase of the modification effort and personal support to the HCW.

### 2.2. Study Design

A retrospective observational study was conducted on a set of objective and economic indicators on the casuistry of HCWs included in the WDMP between July 2017 and December 2019. The data used for the survey were taken from the monthly reports of the WG meetings. The development of a database allowed to trace the path of all the HCWs included in the WDMP. As an indicator of the program efficacy, we measured illness-related absences recorded during six months and one year before, six months and one year after participation in the WDMP in order to define an objective assessment of the path according to the criteria used in WDMP evaluation studies [11,13,16,20,27,29,30,81] and in the promotion paths of health and well-being [78,79]. Such measurement was compared with the company average absenteeism assessed in the same period taken into consideration for the WDMP evaluation. Finally, an assessment was made on the number of limitations among HCWs compared to the national benchmark [82].

### 2.3. Setting and Population in Study and Enrolment

The study was carried out in a large research pediatric hospital with more than 600 settled-bed, located in Rome, Italy, accredited since 2007 by the International Joint Commission. The hospital is regarded as the reference pediatric hospital in central and southern Italy. The total number of ward admissions in 2019 was 29,432. Additionally, more than one million and a half clinical procedures were performed in the outpatient clinics. Finally, about 89,000 patients were admitted to the Emergency and First Aid Department. In the hospital HCWs are about 3300 with a large prevalence of nurses and females. During the period considered, those with problems of internal or external psychological pathologies connected with work related stress, musculoskeletal pathologies or other diseases at risk for limitations and/or disabilities were included in the WDMP.

The participation in the program is voluntary and starts from the request of the HCW or after the proposal of the occupational physician who can recognize the eligibility of the HCW for the program. The percentage of refusals to undertake the program, as well as the percentage of abandonments, is very low (1–2%), because the enrolment process takes place through a very accurate evaluation phase carried out with the participation of the HCW concerned and his/her supervisors.

### 2.4. Economic Analysis

In order to evaluate the costs associated with the days of absence, using the information provided by the hospital’s HR Department, the average cost of the working day was calculated to define the direct costs of the day of absence of each HCW, which is equal to 169.80 Euros. By multiplying the average cost of the day of absence by the number of days saved after six months and one year from the end of the program, the total savings determined by obtained with the program have been defined. The economic evaluation of the one-year total absenteeism savings was also used for the definition of ROI [83] as gross profit in relation to the measurement of the investment cost made by the hospital for the management of WDMP. In this regard, use was made of the average hourly cost of the members of the WG who managed the program, consisting of three occupational medical managers, two administrative managers and one nurse manager multiplied by the total number of hours dedicated to the specific activity. The net profit data relating to the value of savings on days of absence one year after the WDMP conclusion and the investment cost were also used to carry out a BEA, a simulation that allows to identify the volume of activity that represents a balance between costs and revenues [84].

As impact measurement of the WDMP economic effectiveness, the value connected to the productivity recovery of HCWs who undertook the WDMP process, defined in terms of average per-capita annual productivity, was also evaluated. For this computation, on the basis of the calculation of the annual average value of the per-capita productivity at national level according to ISTAT methodology [85], the average annual per-capita productivity of the hospital was defined first by dividing the net profit of the hospital calculated according to the balance in 2019 compared to the number of HCWs (average per-capita annual gross productivity). Secondly, the value of the average per-capita annual productivity of the HCWs included in the WDMP was calculated for the year preceding the program which was burdened by an average absenteeism of 41.5 days (average per-capita annual productivity pre WDMP). From the difference in the average annual per-capita gross productivity (of the hospital) and the average per-capita annual productivity before WDMP, the value of the recovery of average per-capita productivity post WDMP was calculated. This value multiplied by the number of HCWs (84) for whom absenteeism could be calculated one year before and after the WDMP as at 31 December 2019, provides the net profit relative to the recovery of productivity of the personnel who followed the WDMP. The average per-capita annual net productivity recovered after the WDMP has finally been applied to the number of all the HCWs (131) included in the program for the calculation of the value of the total annual productivity recovery.

### 2.5. Study Variables and Indicators

Variables and indicators considered by the study were:Socio-demographic and work-related variables related to the study population (gender, age, occupational category, seniority);Management variables related to the program carried out (WDMP duration, number of evaluations, risk category, typology of intervention implemented;Administrative variables: number of illness-related days of absence in the six months and one year prior to the DM program and in the six months and one year after its conclusion; number of limitations to work in HCWs compared to national data.

### 2.6. Statistical Analysis

A descriptive analysis was carried out in order to define the characteristics of the population studied. For comparisons of means of pre-post scores, a Student paired t-test has been used for normally distributed variables, whereas the Mann Whitney U test has been used for non-normally distributed variables. Chi-square test was performed between qualitative variables. Two-tailed *p*-value <0.05 was considered statistically significant. Data were analyzed using IBM Corp. Released 2017. IBM SPSS Statistics for Windows, Version 25.0. Armonk, NY, USA: IBM Corp.

### 2.7. Ethical Aspects

Our study follows the principles of the Declaration of Helsinki. According to the guidelines on Italian observational retrospective studies, as established by the Italian legislation on the obligatory occupational surveillance and privacy management, HCWs confidentiality was safeguarded and the informed consent was obtained from all the participants regarding the program and the data collected.

## 3. Results

### 3.1. Descriptive Analysis of the WDMP

A total of 131 HCWs was included in the WDMP between July 2017 and December 2019 (equal to about 4% of the hospital personnel), of which 89.7% women, with an average age of 50.4 years (SD ± 8.99) and an average working seniority of 25.3 years (SD ± 12.09) (Table 1). The total number of assessments carried out in the observation period was 901 (219 in 2017, 339 in 2018 and 343 in 2019) with an average number of assessments per individual of 4.8. Out of the total HCWs, 77.8% are nurses. The average duration of the program per HCW was 6.5 months (SD ± 5.3, range 1–29).

In accordance with the case studies relative to occupational disease complaints, the ones examined concerned three categories:Work and extra-work discomfort, mainly of a psychological nature, linked to conditions of work-related stress (29.8% of cases);Musculoskeletal pathology due to biomechanical overload in personnel exposed to the risk of MPM (35.1%);Other problems related to the presence of serious pathologies, especially neoplasms (35.1%).

The type of interventions implemented for the HCWs involved was:Internal reorganization of work duties, keeping destination and job assignment unchanged (tool: interviews with personnel responsible for the area in order to relieve HCWs from activities too heavy for their pathology): 31.7%Proposal to the company management for environmental, ergonomic, structural and technological improvement measures: 24%Paths of psychological support and/or health promotion interventions: 16.4%Change of working destination or, ultimately, change of job: 27.9%.

### 3.2. Analysis of WDMP Impact on Suitability with Limitations of HCWs Compared to the National Benchmark

A first evaluation of the program effectiveness was carried out by verifying the number of job limitations registered among the hospital HCWs from the systematic monitoring of health surveillance data collected between 2017 and 2019 after the WDMP introduction. The job limitations are lower (5.5% of personnel) compared to the national benchmark relative to healthcare facilities (11.8%).

### 3.3. Analysis of WDMP Impact on Illness-Related Absenteeism and Its Economic Evaluation

In those HCWs for whom it was possible to make an assessment of the absences in relation to the end date of WDMP (112 for the calculation within 6 months, 84 for the calculation within 1 year), the days of illness-related absences decreased by 73.3% (*p* < 0.001) in the 6 months following the final date of WDMP compared to the 6 months preceding the start of the program (total number of days of illness related absence in the 6 months before WDMP 2608; post 696) and 66.6% in the course of a subsequent year (*p* < 0.001) (nr. of total days of illness related absence in a year before WDMP 3785; post 1265). The estimated savings associated with the reduction of sick leave absence due to the WDMP are a total of 324,657.60€ over the six months following the conclusion of WDMP and 427,896€ over a year with a per-capita saving of 2898.73€ at 6 months and 4975.53€ at one year (Table 2).

The six-month absenteeism rate among HCWs enrolled in WDMP ranged from 23.3 to 6.2 days (−17.1 days, −73.4%) and one-year rate from 41.5 days to 14.6 days (−26.9 days, −66.6%). These data compared with the company absenteeism rate calculated for the same time frame (6 months, 1 year) and the same periods of time (average of absenteeism between 2017 and 2019 on an annual and half-yearly basis equal to 10.5 and 5.2 days) show after WDMP an absenteeism rate slightly higher than average company rates (Table 3).

The stratification of absenteeism for the variables considered in the study shows a higher reduction of absences after participation in the WDMP of HCWs subdivided into the following categories:males (−79.3%)older age group 51–65 (−71.5%)technicians (−88.3%)belonging to the Specialist Pediatrics Department (−89.5%)affected by chronic-degenerative diseases (−79.6%)recipients of DM interventions that have foreseen the internal reorganization of the work tasks, keeping destination and work task unchanged combined with environmental, ergonomic, structural and technological improvement measures (−98.3%)

### 3.4. Investment Costs for Managing WDMP

The investment cost calculated on the total working hours of the company WG described in the Methods, Section 2, in charge of the evaluations, implementation and monitoring of WDMP interventions was 14,930.77€.

### 3.5. ROI and BEA

On the basis of the investment cost and gross profit obtained by calculating the savings generated by the reduction in days of absence after 1 year from the conclusion of WDMP, a ROI of 27.66€ was calculated for each euro invested. As for BEA, the break-even point is reached by implementing the program at 3.27 HCWs (Table 4).

### 3.6. Analysis of WDMP Impact on Recovery of HCWs Full Productivity

The recovery value in terms of average per-capita annual productivity of the 84 HCWs who completed the WDMP is 12,183.89€ for a total value of recovered productivity for the hospital of 1,023,446.26€ (Figure 1). Assuming to extend the DM program to the cohort of 131 WCWs who participated in the WDMP, the productivity losses generated would be reduced from 2,468,485.10€ to 872,396.34€, with a recovery in value generated by the productivity equal to 1,596,088.76€.

## 4. Discussion

The study has allowed to insert the hospital WDMP among the programs reported in literature that provide for interventions centered on a preventive approach, management of disabilities and permanence at work of HCWs both with limitations (SAW) and on RTW. The case studies examined (131 HCWs, about 900 assessments carried out over the past two and a half years) confirms a higher impact of musculoskeletal problematic conditions and connected to other pathologies, especially neoplastic, in a working population with a high average age and mainly characterized by female subjects and with a qualification as nurse. The study records also the importance of the management and prevention of psychological discomforts of HCWs determined by work related stress in a healthcare setting. The program impact highlights its effectiveness in terms of a statistically significant reduction in illness-related absences and work limitations compared to the company and national data.

The economic evaluation of the program, not so frequently reported in the literature on the evaluation of similar programs [21,22,80,81], provides even more in detail the dimension of the analyzed WDMP effectiveness. ROI and the saving value in the illness-related absences that are high and, even more, the recovery value of the worker productivity, represent the effective size of the impact that a WDMP can have on the human resources of a company, especially at the light of a particularly low and advantageous investment cost and BEA. This aspect is of primary importance for what concerns a health and hospital facility where maintenance, health care and promotion of HCWs are an essential element for the assistance activity connected to the organization mission.

The significant amount of the detected recovery in productivity appears to be connected to the high competence and value of the workforce of a health reality, represented in the study by the higher annual average per-capita gross productivity of the hospital compared to the parameters provided by Istituto Nazionale di Statistica (ISTAT) on a national basis [85]. The value of this difference is also represented by the difficulty of replacing human resources in the healthcare context. This is in fact mainly related to the presence of resources of high intellectual value, such as highly qualified healthcare personnel who might be difficult to be replaced in case of limitation or disability. It should be considered that the organizational structure of the WDMP consisting of a WG, is placed among the functions of the Health Directorate personnel and requires the coordination of the occupational physician, a figure who within the company has a full knowledge of the HCWs’ state of health and the areas of highest risk of the hospital stratified on three levels of intensity (low, medium and high). To face them, the occupational physician, in collaboration with the Health Directorate, represents the competence capable of assessing the areas of the hospital most suitable to accept possible HCWs who need to be relocated. This evaluation can be represented by an indicator of organizational flexibility capable of defining the structure ability to manage and cancel the impact of workers’ limitations or disabilities, favoring their complete recovery at the best possible level of health and, consequently, of productivity. Beyond the main interest in improving health and quality of life of each single worker, the impact of a WDMP, in fact, can be measured on the produced advantages for the economy, society and company in terms of return to productivity, thus determining different and new perspectives for the research and the knowledge translation [16]. This aspect represents one the best added values of the study, being the analysis of the return to productivity a fundamental indicator of the success of a WDMP [16] not so often measured as for profit [81,86,87] and business cases [88,89,90]. However, this aspect represents an important boost to motivate the employers to promote and to invest in similar programs.

This survey confirms and broadens the studies published in literature, especially in recent systematic reviews, on the classification of WDMP models and their evaluation of efficacy, mainly in terms of proactive case management, changes of workplace or equipment, work design and organization, work environment, stakeholder communication [11,12,13,16]. In this regard, compared to the studies reported in some systematic reviews [11,12,13,14,15,16,20,21,22,23,24,25,26], this analysis highlights efficacy and effectiveness assessment based on two objective indicators (illness-related absences, work restrictions), recommended by the guidelines for the evaluation of programs aimed at promoting health and working well-being conducted in workplace settings [78,79]. Unlike most WDMPs that highlight days of sickness higher than a fixed number [13], WDMP analyzed in this study provides for other access criteria, such as a report by the head of the operating unit and the HCW itself or by the system of promotion programs for well-being regarding the presence of HCWs with problems of work-related stress and, in any case, from the entire network of interventions of the integrated WHP plan coordinated by the competent physicians of the Health Department and Occupational Medicine Service.

The program is in fact included in a network of structured services and interventions within the hospital organization as part of a multi-year WHP plan, governed by a set of WGs interconnected and collaborating with each other with the objective of guaranteeing and promoting interventions for the health and well-being of HCWs, who have showed the particular convenience of the adopted prevention strategies [91,92,93,94]. The WGs have, among the others, the task of early highlighting conditions of discomfort and/or pathology, especially with regard to the musculoskeletal or chronic-degenerative system (mainly neoplasms) which require a personalized management with respect to the verification of the working capacity and any adaptation of the working conditions. This aspect represents the main strength of this study as it is based on an integrated program aimed at both prevention and intervention in the event of clear limitation or disability conditions and on an assessment based on objective indicators.

Compared to this proactive approach to early identification and as much as possible based on the prevention of the condition of disability and removal from work, it is evident that a late approach, such as the one activated exclusively by reporting long periods of absence and activation of RTW, does not allow any risk prevention and minimization intervention. Moreover, very often the programs in literature defined as preventive, in practice are applied during illness-related absences and not in the previous phase [25,43,44,45,46,49,51,52].

In this first experience, a top-down approach was preferred for some advantages (speed of action and homogeneity in the processing of operational proposals) that this type of approach may ensure. However, the authors are aware of the usefulness of a participatory, bottom-up approach, which may help to collect and analyze the contribution of workers and can help achieve more stable results over time [95,96,97]. Authors with a proper experience in participatory initiatives [98,99,100,101,102] were included in the research group, with the specific purpose of adopting this type of approach in future disability management activities.

A specific strength of the study is that it has been conducted in a health facility where rarely similar analyses had been conducted. This study presents a few limitations related to some conditions not measured in the analysis conducted. Mainly, the impossibility of referring to the measurement, as recommended in literature, although there are few studies that report data first on job satisfaction of WCWS’ after being included in the WDMP and of the service manager, as well as measurement of physical and mental health indicators before and after the intervention.

Only a part of the workers who were the subject of this analysis could be studied with questionnaires that measure quality of life and psychological well-being [(The Sherbourne Quality of Life Assessment Questionnaire (SF-36) and the Goldberg’s General Health Questionnaire (GHQ12)] at the beginning and end of the observation period, that showed a very significant improvement of their general health and quality of life. A desirable development of this study may consider an analysis of the variations of these health parameters and their association with treatment.

Beyond the indirect measurement of these conditions connected to the conclusion of the path proposed by the WG and approved by the HCW, measurement of subjective parameters of health and working well-being and connected to job satisfaction, can be implemented within the analyzed WDMP to verify, also from a subjective point of view, the positive conclusion of the path.

Moreover, this study was a retrospective analysis of records from a company-sponsored proprietary program to improve RTW rates and this may undoubtedly bias the results. Furthermore, for ethical reasons the offer of disability program has been extended to all the workers of the company and this prevents having a control group. However, the favorable results obtained in the area of disability management provide a reasonable starting point for extending the experience to other companies; in particular, the collaboration of the doctors of the Catholic University and the Gemelli Polyclinic will allow the realization of multicentric studies in the future, which will be able to overcome the problems present in this experience and will provide a non-randomized analysis of pre- and post-outcomes.

## 5. Conclusions

The evaluation of the program carried out through this study seems to demonstrate the effectiveness of the implemented WDMP model, acting positively on the variables that influence productivity (absenteeism, disability, chronic diseases, workforce ageing) and allowing to restrict limitations within the hospital.

The originality of our study consists in the use and connection of different methods of analysis for the evaluation of the program effectiveness and efficiency. The analysis connects, in fact, through the use of two objectives indicators (absenteeism and limitations) with a complex economic analysis conducted through the use of several parameters (ROI, BEA and return to productivity) used separately in other studies reported in the literature.

The study contributes to deepen the knowledge on the importance of implementing an integrated WDMP in a specific structure, such as a hospital, where health promotion and wellbeing of HCWs constitute a critical issue. The study mainly aims at demonstrating the program effectiveness in terms of reduction of absenteeism and limitations of the hospital personnel and the efficiency of the investment defining the power of the experience. Moreover, the definition of return to productivity of the HCWs involved represents the added value of the study, providing the right dimension of the power of such a preventive approach to disability and aging of the working population.

The experience carried out over the years within the hospital will lead to the development of a protocol, structured according to the WDMP model, with the management’s consent as a “company good practice.” Redefinition and recoding of the key steps of the DM process and evaluation of the results achieved, also through the analysis of key indicators, such as pre-post-illness-related absences, is contributing to the implementation of the model. This will allow a review of the WDMP model to be proposed within the Italian healthcare organizations. A further development of the protocol should include the possible introduction of the professional profile of the Disability Manager as responsible for the entire program.

In future multicenter investigations, a qualitative analysis may be included, seeking to understand the reasons, arguments and attitudes relevant to understanding the factors that facilitate productivity, health and well-being of employees.

## Figures and Tables

**Figure 1 ijerph-17-08084-f001:**
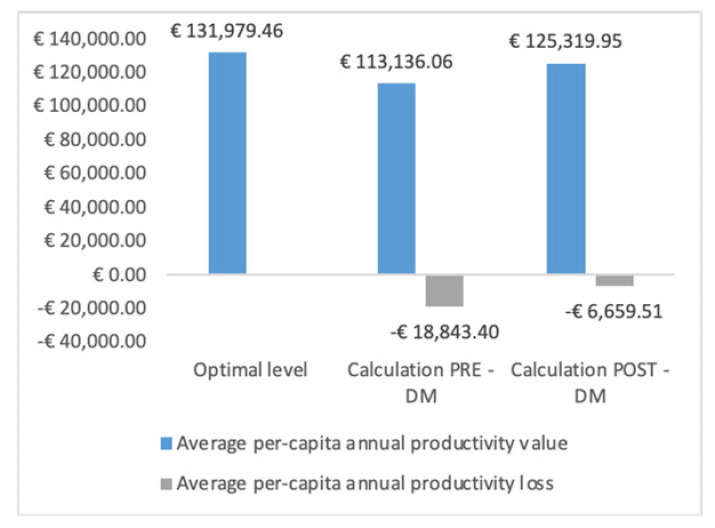
Value of the average annual net productivity recovered.

**Table 1 ijerph-17-08084-t001:** Workplace Disability Management Program (WDMP) case studies (2017–2019).

Cases Treated (2017–2018)	Average Age of Seniority (DS)	Profession	Average Number of Evaluations per Healthcare Worker (HCW)	Type of Case Studies
131 (89.7% females)	50.4 (±8.99) 25.3 (±12.09)	77.8% Nurse	4.8	Discomfort 29.8% Musculoskeletal pathologies 35.1% Severe pathologies 35.1%

**Table 2 ijerph-17-08084-t002:** Economic evaluation of illness related days of absence before and after WDMP.

Days of Absence 6 Months Pre and Post WDMP	Days of Absence 1 Year Pre and Post WDMP	Valorization of Total Savin gof Absences	Per-Capita Cost Reduction
Pre = 2608 gg Post = 696 gg (−73.3%, *p* < 0.001)	Pre = 3785 Post = 1265 (−66.6%, *p* < 0.001)	At 6 months 324,657.6€ At 1 year 427,896€	At 6 months 2898.73€ At 1 year 4975.53€

**Table 3 ijerph-17-08084-t003:** Comparison of absenteeism rates at six months and one year before and after WDMP, as well as company absenteeism.

Absenteeism Rate 6 Months Pre- and Post-(days) WDMP (∆, ∆ % *p*-Value)	Company Half-Yearly Absenteeism Rate (days/semester)	Absenteeism Rate 1 Year Pre- and Post- (days) WDMP (∆, ∆ % *p*-value)	Company Yearly Absenteeism Rate (days/year)
Pre 23.3 (IC95% 15.99–30.61) Post 6.2 (IC95% 4.30–8.10) −17.1gg −73.4%	5.2	Pre 41.5 (IC95% 28.28–54.92) Post 14.6 (IC95% 9.39–20.01) −26.9gg −66.6%	10.5

**Table 4 ijerph-17-08084-t004:** Return on Investment (ROI) and Break-Even Analysis (BEA) of the WDMP.

Economic Evaluation of Total Saving of Absences (After 1 Year)	Investment Cost	Return on Investment (ROI) for Each Euro	Break-Even Analysis (BEA) (nr. HCWs)
427,896€	14,930.77€	€ 27.66€	3.27

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
