# Peer review of "Return on Investment (ROI) and Development of a Workplace Disability Management Program in a Hospital—A Pilot Evaluation Study"

_ijerph, 2020, doi:10.3390/ijerph17218084_

Round 1
Reviewer 1 Report
The article presents a current and relevant topic for the study of employee motivation and in programs aimed at preventing the variables that influence productivity (absenteeism, disability, chronic diseases, aging of the workforce).
Strengths: literature review, methodology, discussion of results
Weaknesses: Study of a hospital organization, conclusions.
Suggestions for improvement: The main conclusions of the study could be systematized in detail, indicating the originality and value of the study.
As future investigations, a qualitative analysis may be included, seeking to understand the reasons, arguments and attitudes relevant to understanding the factors that facilitate productivity, health and well-being of employees.
Author Response
The article presents a current and relevant topic for the study of employee motivation and in programs aimed at preventing the variables that influence productivity (absenteeism, disability, chronic diseases, aging of the workforce). Strengths: literature review, methodology, discussion of results. Weaknesses: Study of a hospital organization, conclusions.
Suggestions for improvement: The main conclusions of the study could be systematized in detail, indicating the originality and value of the study.
As future investigations, a qualitative analysis may be included, seeking to understand the reasons, arguments and attitudes relevant to understanding the factors that facilitate productivity, health and well-being of employees.
Response: Thank you for your valuable observations and suggestions that allowed us to improve our manuscript. We agree with your suggestions to ameliorate the conclusions of the study. We added one paragraph at lines 470-482:
The originality of our study consists in the use and connection of different methods of analysis for the evaluation of the program effectiveness and efficiency. The analysis connects, in fact, through the use of two objectives indicators(absenteeism and limitations) with a complex economic analysis conducted through the use of several parameters (ROI, BEA and return to productivity) used separately in other studies reported in literature.
The study contributes to deepen the knowledge on the importance of implementing an integrated WDMP in a specific structure, such as a hospital, where health promotion and wellbeing of HCWs constitute a critical issue. The study mainly aims at demonstrating the program effectiveness in terms of reduction of absenteeism and limitations of the hospital personnel and the efficiency of the investment defining the power of the experience. Moreover, the definition of return to productivity of the HCWs involved represents the added value of the study, providing the right dimension of the power of such a preventive approach to disability and aging of the working population.
Another paragraph was added at Lines and 492-494:
In future multicentre investigations, a qualitative analysis may be included, seeking to understand the reasons, arguments and attitudes relevant to understanding the factors that facilitate productivity, health and well-being of employees.
Reviewer 2 Report
Manuscript Review, Int J Environ Res Public Health, 13 Oct 2020
“Return on investment (ROI) and development of a Workplace Disability Management Program in a hospital. A pilot evaluation study”
This manuscript summarizes return-to-work (RTW) outcomes for 131 healthcare workers participating in a disability management program instituted by a children’s hospital in Italy. The primary analytic strategy is to compare RTW outcomes of participants with the RTW rates reported in prior years and to calculate a return on investment (ROI) in terms of relative savings vs. program costs. The primary result was a ROI of 27.66 euros per euro spent, and based on these results, the program would have reached a break-even point after 3.27 workers were exposed to the program. The authors conclude evidence that their implemented DM program acted positively on the variables that affect productivity and work limitations.
This study is a retrospective analysis of records from an industry-sponsored proprietary program to improve RTW rates in the authors’ home institution, and therefore the conclusions of the program do not provide the unbiased conclusions that might be generated from a randomized trial or researcher-initiated study. However, there is a place for well-written and detailed industry reports of successful programs in the area of disability management, and this provides a reasonable data source to provide a non-randomized analysis of pre- and post-outcomes. Detailed comments follow:
- One limitation of the study is that it was entirely crafted by managers within the organization; there are no participatory elements described in the planning and implementation of the program. Can this be addressed? Is there any evidence of program elements that represented a joint effort between labor and management?
- A thorough proofread by a native English speaker would be helpful to improve the flow of ideas and correct some awkward phrasing.
- A better operational description of the DM program is needed. Most of the existing language describes philosophy and goals of the program without translating these into specific procedures. How would a worker receive assistance differently under the new program?
- Discussion: The ROI approach presumes an employer perspective to assess economic benefits and effectiveness of the program. Is there any indication from study results that workers experienced improved health and work outcomes (apart from being able to maintain employment)?
- Discussion: The paragraphs on strengths and limitations should be rewritten to focus on potential sources of bias in the study design and data collection (vs. a randomized design).
- Methods: The lack of information about sample sizes in the pre-DM sample make it difficult to assess the statistical strengths and limitations.
- Ethical aspects: The manuscript states that informed consent was obtained from all participants, but it’s unclear what workers were agreeing to. Was this consent to participate in the DM program or to use their outcome data or both?
- Enrollment: It’s unclear whether the DM program was voluntary for temporarily disabled workers. If the program was voluntary, what percentage of workers agreed to participate? This has implications in terms of validity of pre-post comparisons and generalizability of results.
- Supervisor involvement: To what extent were supervisors within wards and departments included in the job modification effort being championed by the occupational medicine department?
Author Response
This manuscript summarizes return-to-work (RTW) outcomes for 131 healthcare workers participating in a disability management program instituted by a children’s hospital in Italy. The primary analytic strategy is to compare RTW outcomes of participants with the RTW rates reported in prior years and to calculate a return on investment (ROI) in terms of relative savings vs. program costs. The primary result was a ROI of 27.66 euros per euro spent, and based on these results, the program would have reached a break-even point after 3.27 workers were exposed to the program. The authors conclude evidence that their implemented DM program acted positively on the variables that affect productivity and work limitations.
This study is a retrospective analysis of records from an industry-sponsored proprietary program to improve RTW rates in the authors’ home institution, and therefore the conclusions of the program do not provide the unbiased conclusions that might be generated from a randomized trial or researcher-initiated study. However, there is a place for well-written and detailed industry reports of successful programs in the area of disability management, and this provides a reasonable data source to provide a non-randomized analysis of pre- and post-outcomes. Detailed comments follow:
1) One limitation of the study is that it was entirely crafted by managers within the organization; there are no participatory elements described in the planning and implementation of the program. Can this be addressed? Is there any evidence of program elements that represented a joint effort between labor and management?
Response: This pilot study, for the first time, implemented a disability management program in a hospital.. The top-down approach was preferred in the first application phase, as it provides speed and homogeneity in processing the operational proposals that this type of approach ensures. However, the authors are aware of the usefulness of a bottom-up approach, which collects and analyzes the workers’ contribution.
Authors with proper experience in participatory initiatives ] were included in the research group, with the specific purpose of adopting this type of approach in future disability management activities [Magnavita N. Medical Surveillance, Continuous Health Promotion and a Participatory Intervention in a Small Company. Int J Environ Res Public Health. 2018 Apr 2;15(4). pii: E662. doi: 10.3390/ijerph15040662 --- Magnavita N., Santoro PE, Rossi I, Bergamaschi A. Validation and use of the P.A.D. questionnaire for the participatory definition of a policy to combat addictions in the workplace G. Ital Med Lav Ergon 2009; 3 (Supp.2):237-238 --- Magnavita N, Magnavita G, Bergamaschi A. Definition of participatory policy on alcohol and drug in two health care companies. G Ital Med Lav Ergon 2010; 4 (Suppl 2): 300-301 --- Magnavita N. Aging workforce. The importance of work engagement and participatory ergonomics. HPNCDs Health Policy in NonCommunicable Diseases. 2016; 3:56-65].
A specific paragraph was added at lines 428-434.
2) A thorough proofread by a native English speaker would be helpful to improve the flow of ideas and correct some awkward phrasing.
Thank you for the request. A revision of the article was made by a native English speaker in order to simplify the parts more difficult to understand.
3) A better operational description of the DM program is needed. Most of the existing language describes philosophy and goals of the program without translating these into specific procedures. How would a worker receive assistance differently under the new program?
We agree with your observation that allowed us to better describe our program.
There are three main reasons for entering the HCWs in the program: intra or extra work psychological distress (work related stress, family problems); musculoskeletal disorders; other pathologies, especially neoplastic diseases. These are the main reasons that impact on the work capacity of the HCW who very often manifests discomfort, an altered health condition and/or is often absent from work. The paths that are proposed by the WG are personalized and mainly concern four types of intervention
- Internal reorganization of work duties, keeping destination and job assignment unchanged
- Proposal to the company management for environmental, ergonomic, structural and technological improvement measures
- Paths of psychological support and/or health promotion interventions
- Change of working destination or, ultimately, change of job
The assessment of each case is carried out during the WG meetings and the decisions on the adequate intervention to be implemented or changed along the way are discussed and agreed with the HCW and his/her supervisors. The path can last several months, on average. It is closed when the criticalities detected at the entrance have found the appropriate solution. The various steps of the path are ratified within the monthly WG meetings.
This description is inserted in the text at lines 146-152.
4) Discussion: The ROI approach presumes an employer perspective to assess economic benefits and effectiveness of the program. Is there any indication from study results that workers experienced improved health and work outcomes (apart from being able to maintain employment)?
We are grateful to the reviewer who made highlighted a relevant issue. In some of the cases reported in this study, workers were asked to complete two questionnaires that measure quality of life and psychological well-being (The Sherbourne Quality of Life Assessment Questionnaire (SF-36), and the Goldberg's General Health Questionnaire (GHQ12) at the beginning and end of the observation period,that showed a very significant improvement. The detailed analysis of the variations and the association of the variations with some treatment parameters will be the subject of a publication that we plan to produce in the near future.
A sentence was added at lines 443-448.
5) Discussion: The paragraphs on strengths and limitations should be rewritten to focus on potential sources of bias in the study design and data collection (vs. a randomized design).
We have added these considerations in the "strengths and weaknesses" section. Our study was a retrospective analysis of records from a company-sponsored proprietary program to improve RTW rates, and this may undoubtedly bias the results. Furthermore, for ethical reasons the offer of disability program has been extended to all the workers of the company, and this prevents having a control group. However, the favourable results obtained in the area of disability management provide a reasonable starting point for extending the experience to other companies; in particular, the collaboration of the physicians of Catholic University and Gemelli Polyclinic will allow to carry out in the future multicentric studies, which will help overcome the problems present arisen in this experience and provide a non-randomized analysis of pre- and post-outcomes
These observations were added at lines 455-463
6) Methods: The lack of information about sample sizes in the pre-DM sample make it difficult to assess the statistical strengths and limitations.
The population of workers potentially eligible for the program relates to all hospital HCWs (about 3000). The comparison made for sick leave before and after the program is therefore made with the entire population of the hospital's HCWs. (See table no. 3 in the article)
7) Ethical aspects: The manuscript states that informed consent was obtained from all participants, but it’s unclear what workers were agreeing to. Was this consent to participate in the DM program or to use their outcome data or both?
According to the legislation in force in Italy, all hospital HCWs sign a consent during the health surveillance activities, which they must obligatorily submit, in which it is specified that all the data collected regarding their health conditions and activities and programs carried out for prevention and health promotion can be used for epidemiological research. In this case, therefore, the consent is related to participation in the WDMP and, consequently, to the data collected during the related activities.
A brief clarification has been added at lines 258.
8) Enrollment: It’s unclear whether the DM program was voluntary for temporarily disabled workers. If the program was voluntary, what percentage of workers agreed to participate? This has implications in terms of validity of pre-post comparisons and generalizability of results.
The participation in the program is voluntary and starts from the request of the HCW or after the proposal of the occupational physician who can recognize the eligibility of the HCW for the program. The percentage of refusals to undertake the program, as well as the percentage of abandonments, is very low (1-2%), because the enrolment process takes place through a very accurate evaluation phase carried out with the participation of the HCW concerned and his/her supervisors.
The description of these aspects was inserted in the text at lines 192-197.
9) Supervisor involvement: To what extent were supervisors within wards and departments included in the job modification effort being championed by the occupational medicine department?
The supervisors and staff responsible of the wards and departments were involved in all the phases of the program from the evaluation of the case to the decision regarding the objective of the personal program of the HCWs and, necessarily in the phase of the modification effort and personal support to the HCW.
A specific paragraph was added at lines 163-166.